# Blending Concepts in Text-to-Image Diffusion Models using the Black Scholes Algorithm

## Abstract

Many image generation tasks, such as content creation, editing, personalization, and zero-shot generation, require generating unseen concepts without retraining the model or collecting additional data. These tasks often involve blending existing concepts by conditioning the diffusion model with text prompts at each denoising step, a process known as "prompt mixing". We introduce a novel approach for prompt mixing that forecasts predictions regarding the generated image and makes informed text conditioning decisions at each diffusion step. By leveraging the connection between diffusion models, rooted in non-equilibrium thermodynamics, and the Black-Scholes model for pricing options in finance, we derive an appropriate algorithm for prompt mixing. Specifically, the parallels between diffusion models and the Black-Scholes model enable us to leverage properties related to the dynamics of the Markovian model derived in the Black-Scholes algorithm. Our prompt-mixing algorithm is data-efficient, requiring no additional training, and operates without human intervention or hyperparameter tuning. We highlight the benefits of our approach by comparing it, both qualitatively and quantitatively using CLIP scores, to other prompt mixing techniques. These include linear interpolation, alternating prompts, step-wise prompt switching, and CLIP-guided prompt selection across various scenarios such as single object per text prompt, multiple objects per text prompt, and objects against backgrounds. Code will be made publicly available.

## 1 Introduction

Text-to-image diffusion models Ho et al. (2020; 2022) often need to generate new and unseen concepts for tasks such as content creation, editing, personalization, and zero-shot generation without retraining the model or collecting additional task-specific data. While advanced versions of these models attribute their improved capabilities to factors like increased training data, enhanced text understanding networks, and improved image generation architectures, data-efficient solutions Wang et al. (2024) leveraging pre-trained diffusion models can enhance results without requiring additional data collection or increasing model complexity.

Often, the desired results can be formulated at the intersection of various individual manifolds familiar to the model. This can be seen as blending existing concepts by conditioning the diffusion model using text prompts at each denoising step, a process known as "prompt mixing" Patashnik et al. (2023); Balaji et al. (2022); Repo (2023); Pinkney (2024). For instance, generating an image resembling a hybrid of a pink cat and a dog. While current diffusion models Adobe (2024); Saharia et al. (2022); Betker et al. (2023); Esser et al. (2024) excel at combining individual text prompts to create visually appealing combinations in simple scenarios, these capabilities are often restricted to advanced model versions and rely on large-data techniques.

Data-efficient prompt mixing techniques can enhance image generation without additional data. While linear interpolation Kawar et al. (2023) of text embeddings from individual prompts is a simple approach, it may not be optimal due to the highly non-linear nature of the text-image manifold and potential bias issues. Alternatives include switching between prompts during diffusion denoising, either by alternating Kothandaraman et al. (2023a) or using a step-wise technique Patashnik et al. (2023). However, these methods often require human involvement and careful prompt engineering for optimal results.

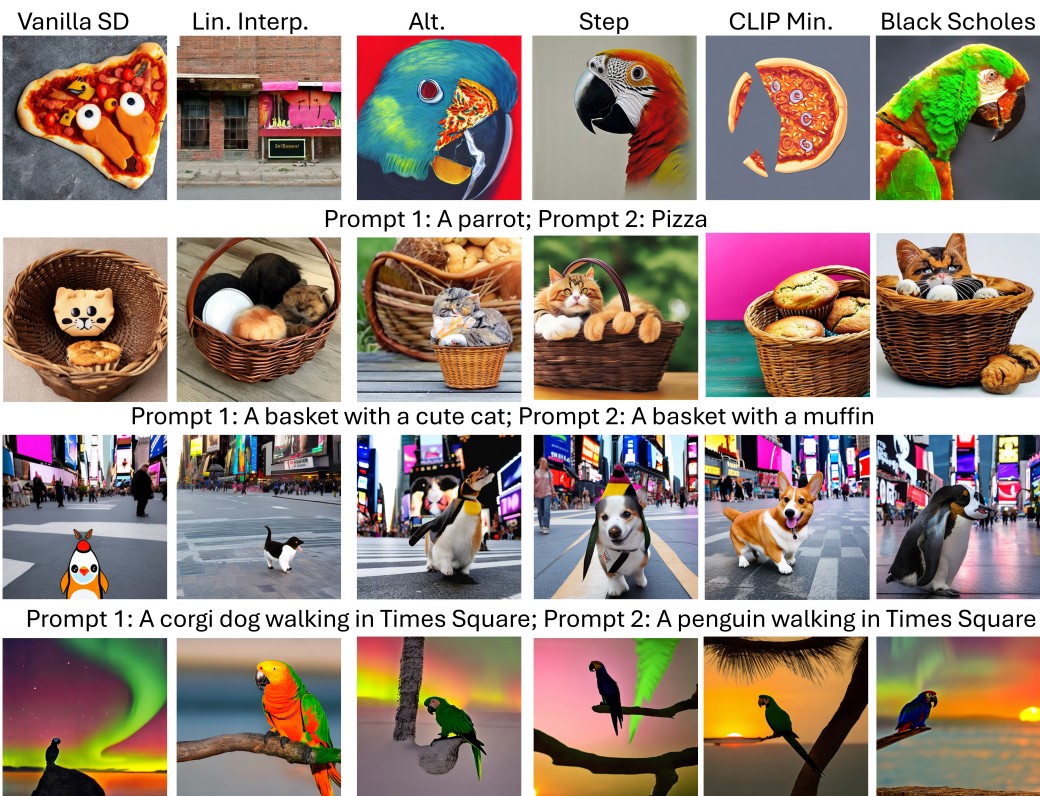

Vanilla SD    Lin. Interp.    Alt.    Step    CLIP Min.    Black Scholes

Prompt 1: A parrot; Prompt 2: Pizza

Prompt 1: A basket with a cute cat; Prompt 2: A basket with a muffin

Prompt 1: A corgi dog walking in Times Square; Prompt 2: A penguin walking in Times Square

Prompt 1: A parrot watching the sunset; Prompt 2: A parrot watching northern lights

Figure 1: Our method's results (**Black Scholes**, last column) are presented alongside comparisons to prior work. Vanilla stable diffusion (SD) struggles to capture clear characteristics of individual text prompts (notably missing distinct features such as those of the parrot, cat, dog/penguin, and sunset/penguin). Linear interpolation performs poorly due to non-linear manifolds. Alternating sampling and step-wise switching yield low-quality results with artifacts, primarily because they lack intelligent prompt selection during denoising steps (missing characteristics of pizza, artifacts in cat/muffin and dog/penguin mixing, sunset/northern lights not well captured). CLIP-min exhibits bias issues by not modeling diffusion denoising dynamics and prompt selection effectively, which hinders fore-sighted decision making, the generated images are biased towards one of the text prompts. In contrast, our Black-Scholes model generates realistic images that meticulously balance and preserve the characteristics of each individual text prompt. The images are from set 1, set 2, set 3 and set 4 (Refer Section 5.1) respectively.

The question arises: what's the best way to switch between text prompts for prompt-mixing? An automated approach considers the model's varying capabilities with different prompts. While attention maps Hong et al. (2023); Chefer et al. (2023) and layout guidanceZheng et al. (2023); Cheng et al. (2023) methods excel at guiding the model toward distinct scene entities, they may not be as effective for blending concepts within the same entity. To address this, during each step of diffusion denoising, the network should automatically focus on the text prompt that corresponds to the image's deficiencies.

**Main contributions.**    In this paper, we introduce a novel approach to prompt mixing by integrating concepts from economics and finance. Our objective is to leverage pretrained diffusion models for prompt-mixing to generate images at the intersection of various text-image manifolds. During each step of the diffusion denoising process, our method dynamically conditions on the text prompt that requires the highest level of attention. To achieve this, our algorithm assesses the 'cost' associated with each relevant text prompt and selects the optimal conditions for the subsequent step of image generation based on the 'cost' that requires maximum optimization.

**Key Insight:** The inspiration for our approach lies in non-equilibrium thermodynamics Sohl-Dickstein et al. (2015), which is at the core of the development of diffusion models. These models share a conceptual foundation with the Black Scholes modelMerton (1976), a Nobel Prize-winning mathematical framework extensively employed in financial markets for pricing European call optionsBlack & Scholes (2019). The denoising steps within a diffusion model, aimed at image generation, constitute a Markovian time-series. In our analogy, the image itself represents a valuable 'stock' or 'asset' that we aim to 'purchase' (or generate) at the most favorable 'cost' (or alignment with the text prompts). During each step of the diffusion denoising process, we *extrapolate the generate latents to compute the 'stock' prices at the corresponding time-step*, and *use the properties of diffusion models to compute the various variables involved in the Black Scholes algorithm*. Consequently, we *leverage the Black Scholes model to predict a score for each text prompt*. This score serves as *an indicator of how the image should be conditioned in the subsequent timestep*. By adopting this approach, our model dynamically focuses on aspects that require *attention*, ultimately generating an image that optimally satisfies all relevant text prompts.

We perform experiments on prompts with varying complexities to assess our method's ability to seamlessly blend different objects and backgrounds across scenarios. Through CLIP-based quantitative and qualitative comparisons against several baselines such as vanilla stable diffusion, linear interpolation, alternating sampling, prompt switching, and CLIP guided prompt switching, we highlight the superiority of our approach.

## 2 RELATED WORK

**Prompt mixing.** Prompt engineering Witteveen & Andrews (2022) is an effective approach that includes techniques like rephrasing prompts to enhance model generalization. Another method involves using large language models (LLMs) Lian et al. (2023); Wu et al. (2023) to parse complex prompts and identify useful priors for image generation. Additionally, attention maps Chefer et al. (2023) and prompt mixing techniques contribute to achieving better results. Prompt mixing is a technique where different text prompts are used at various steps during the diffusion denoising process. Patashnik et al. Patashnik et al. (2023) follow a step-wise approach, while Aerial Diffusion Kothandaraman et al. (2023a) alternates between prompts to create semantically consistent aerial-view images. Tools such as Image Mixer Diffusion Pinkney (2024) and CLIP Guided Image Mixing Repo (2023) follow similar techniques of prompt switching to blend multiple text prompts. However, these methods often require complex hyperparameter tuning.

**Mixing step.** The mixing time or mixing step Levin & Peres (2017) of a Markov chain refers to the duration it takes for the chain to reach its steady-state distribution. Mixing time has found applications in image editing Zhu et al. (2024) and synthesizing out-of-distribution (OOD) images Zhu et al. (2023). When switching between prompts using mixing time, a step-wise approach is followed, akin to the work by Patashnik et al. Patashnik et al. (2023). The advantage of mixing-time approaches over the method proposed by Patashnik et al. Patashnik et al. (2023) is that it mathematically determines the optimal switching time without requiring complex hyperparameter tuning. To approximate the mixing time, it estimates the radius of the latent space. However, there are a few limitations to these methods. Firstly, most of these approaches are applied to diffusion models trained on relatively small, problem-specific datasets. Secondly, identifying clear boundaries on large-scale foundation models remains an unsolved challenge. Additionally, these methods lack the flexibility to choose the most optimal prompt at each timestep.

**Understanding the latent space.** Tangential to our work, there has been substantial work Karras et al. (2017); Abdal et al. (2019); Gal et al. (2022) on exploring the latent space. These investigations have yielded valuable insights and practical solutions for downstream tasks such as image editing and manipulation applications Zhu et al. (2016); Shen et al. (2020); Kwon et al. (2022); Zhu et al. (2020); Preechakul et al. (2022). Furthermore, a deeper understanding of the latent space within diffusion models has led to advancements in various methods Rombach et al. (2022); Kwon et al. (2022); Yang et al. (2023b).

## 3 THE BLACK SCHOLES ALGORITHM AND DIFFUSION MODELS

### 3.1 THE BLACK SCHOLES MODEL

In this section, we provide a brief overview of the Black Scholes pricing model Merton (1976) used to determine the price of European call options of assets. In simple terms, a European call option allows an investor to lock in the price of an asset at any time but permits stock purchase (if desired) only upon expiration. Regardless of whether the stock price moves favorably or unfavorably over time, this option structure remains consistent. Investors rely on the Black-Scholes model to predict stock prices over time and make informed decisions about the optimal timing for stock purchases. The Black Scholes formula involves 5 key variables:

1. Underlying stock price or spot price $S$: This represents the current price of the asset.

2. Strike price $K$: The strike price is the cost of the asset at the time of expiry.

3. Time to expiration $t$: It measures the time difference between the current moment and the expiry time.

4. Volatility $\sigma$: Volatility reflects the variation in prices of the asset.

5. Risk free rate $r$: The risk-free rate is the minimum return on an investment when the investor faces zero risks.

To obtain the **Black Scholes score** of purchasing an asset, the spot price $S$ is first multiplied by the standard normal probability distribution function. From this result, to obtain the final cost $C$, the strike price $K$ multiplied by the cumulative standard distribution function is subtracted. Mathematically,

$$SN(d_1) - Ke^{-rt}N(d_2), \tag{1}$$

where

$$d_1 = \frac{log\frac{S}{K} + (r + \frac{\sigma^2}{2})(t)}{\sigma\sqrt{t}}, d_2 = d_1 - \sigma\sqrt{t}. \tag{2}$$

### 3.2 RELATION TO DIFFUSION MODELS

#### 3.2.1 DIFFUSION MODELS

The main concept behind diffusion models involves iteratively adding small amounts of random Gaussian noise to transform an initial photorealistic image $x_0$ into noise $x_T \sim \mathcal{N}(0, I)$ over $T$ steps. This process is known as the **forward process**. Conversely, starting from random noise $x_T \sim \mathcal{N}(0, I)$ and refining it iteratively for $T$ steps can generate a photorealistic image $x_0$. Since diffusion is gradual, $T$ is typically large. At each intermediate timestep $t \in \{0, \ldots, T\}$, $x_t$ satisfies:

$$x_t = \sqrt{\alpha_t}x_0 + \sqrt{1 - \alpha_t}\epsilon_t$$

The hyperparameters of the diffusion schedule are $0 = \alpha_T < \alpha_{T-1} < \ldots < \alpha_1 < \alpha_0 = 1$, and $\epsilon_T \sim \mathcal{N}(0, I)$. To obtain $x_{t-1}$ at each refinement step, the neural network $f_\theta(x_t, t)$ is applied along with the corresponding random Gaussian noise perturbation. Essentially, during each step of diffusion denoising, the known variance in added noise follows a Gaussian distribution.

#### 3.2.2 ANALOGIES BASED ON THERMODYNAMICS

In finance, volatility refers to the standard deviation in the way stock prices change over time. The mathematical description of diffusion models, derived from non-equilibrium thermodynamics, shares similarities with the derivation of the Black-Scholes model used in pricing European call options within financial markets. Both models emerge from similar assumptions and conditions, underpinned by a shared mathematical structure.

The process of diffusion can be understood using tools from statistical mechanics, where the generative dynamics undergo phase transitions and symmetry breaking. The dynamic equation of the generative process can be interpreted as a stochastic adiabatic transformation that minimizes free

energy while keeping the system in thermal equilibrium. The Black-Scholes model assumes that the stock price follows a geometric Brownian motion, which can be described by a stochastic differential equation (SDE), where Itô's Lemma is used to derive the partial differential equation (PDE) that describes the option price dynamics. The pricing of derivatives can be seen as a process of minimizing a certain "free energy" under constraints.

In terms of thermodynamic analogies, both models can be interpreted through the lens of thermodynamics, particularly in terms of free energy minimization and equilibrium states. In diffusion models, the reverse diffusion process can be seen as minimizing a free energy functional:

$$F[p] = \int p(\mathbf{x}) \left( \log p(\mathbf{x}) - \log q(\mathbf{x}) \right) d\mathbf{x}$$

where $q(\mathbf{x})$ is the target distribution. In the Black-Scholes model, option pricing can be viewed as a process of minimizing a financial free energy under constraints, analogous to thermodynamic systems seeking equilibrium.

Diffusion models introduce Gaussian noise to data in a controlled manner to learn the underlying data distribution. The Black-Scholes algorithm models the randomness in stock prices using Brownian motion, capturing the inherent uncertainty in financial markets. In diffusion models, the forward process drives the system out of equilibrium by adding noise, while the reverse process aims to bring it back to a structured state. In the context of the Black-Scholes equation, the financial market can be seen as a non-equilibrium system where prices fluctuate, and the option pricing model seeks to find a fair value (equilibrium price) under risk-neutral assumptions.

### 3.2.3 ANALOGIES BASED ON SDEs

Both diffusion models and the Black-Scholes equation rely on SDEs to describe the evolution of systems over time. In diffusion models, the forward diffusion process is described by the SDE:

$$\mathbf{x}_t = -\frac{1}{2}\beta_t \mathbf{x}_t \, dt + \sqrt{\beta_t} \, d\mathbf{w}_t$$

where $\mathbf{x}_t$ is the state at time $t$, $\beta_t$ is a time-dependent noise coefficient, and $\mathbf{w}_t$ is a Wiener process (Brownian motion). The reverse process, which reconstructs the data, is given by:

$$\mathbf{x}_t = \left( \frac{1}{2}\beta_t \mathbf{x}_t + \nabla_{\mathbf{x}_t} \log p_t(\mathbf{x}_t) \right) dt + \sqrt{\beta_t} \, d\mathbf{w}_t$$

where $\nabla_{\mathbf{x}_t} \log p_t(\mathbf{x}_t)$ is the score function. In the Black-Scholes equation, the stock price $S_t$ follows a geometric Brownian motion:

$$dS_t = \mu S_t \, dt + \sigma S_t \, dW_t$$

where $\mu$ is the drift rate, $\sigma$ is the volatility, and $W_t$ is a Wiener process. Using Itô's Lemma, the option price $V(S, t)$ evolves according to:

$$dV = \left( \frac{\partial V}{\partial t} + \mu S \frac{\partial V}{\partial S} + \frac{1}{2}\sigma^2 S^2 \frac{\partial^2 V}{\partial S^2} \right) dt + \sigma S \frac{\partial V}{\partial S} dW_t$$

In terms of Partial Differential Equations (PDEs), both models lead to PDEs that describe the evolution of probability densities or option prices. In diffusion models, the Fokker-Planck equation describes the time evolution of the probability density $p(\mathbf{x}, t)$:

$$\frac{\partial p(\mathbf{x}, t)}{\partial t} = -\nabla \cdot (\mathbf{f}(\mathbf{x}, t) p(\mathbf{x}, t)) + \frac{1}{2}\nabla \cdot (D(\mathbf{x}, t)\nabla p(\mathbf{x}, t)$$

where $\mathbf{f}(\mathbf{x}, t)$ is the drift term and $D(\mathbf{x}, t)$ is the diffusion coefficient. The Black-Scholes PDE for the option price $V(S, t)$ is:

$$\frac{\partial V}{\partial t} + rS\frac{\partial V}{\partial S} + \frac{1}{2}\sigma^2 S^2 \frac{\partial^2 V}{\partial S^2} - rV = 0$$

where $r$ is the risk-free interest rate.

In diffusion models, the reverse process can be described by a PDE that governs the evolution of the probability density function. In the Black-Scholes equation, the option pricing model is derived as a PDE that describes the evolution of the option price over time. Diffusion models use the score function (gradient of the log probability) to guide the reverse diffusion process. In the Black-Scholes equation, the gradient of the option price with respect to the stock price (delta) is crucial for constructing a risk-neutral portfolio.

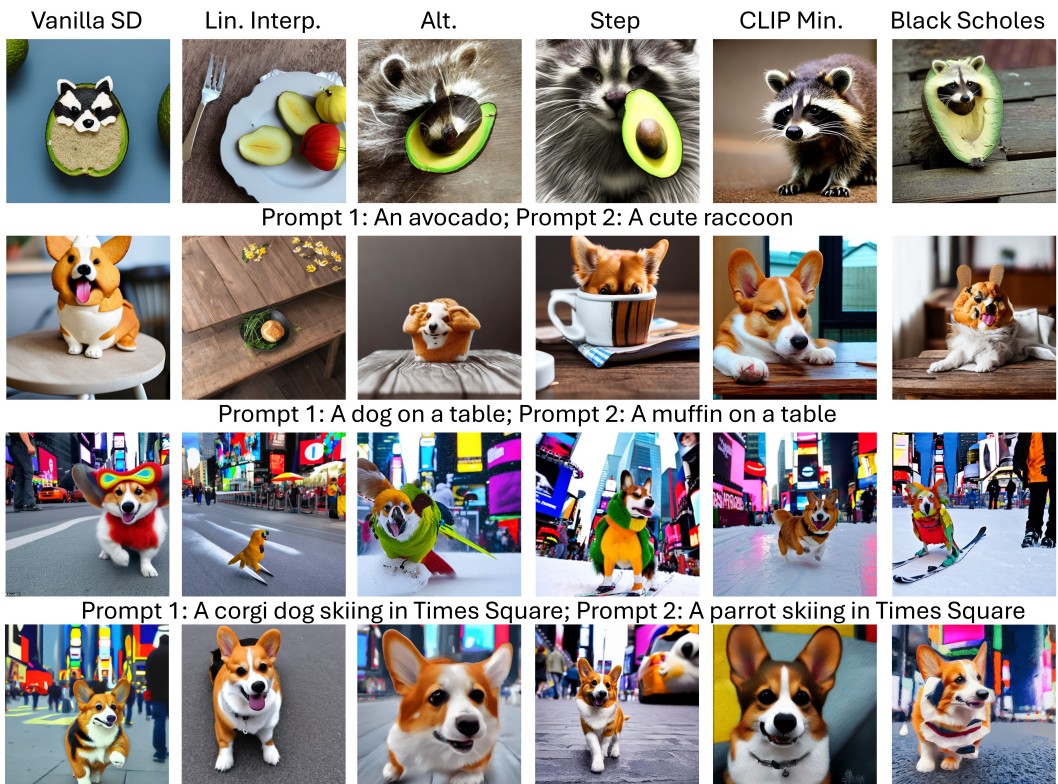

Figure 2: We present more results of our method along with comparisons. Vanilla SD fails to capture clear characteristics of individual text prompts, omitting distinct features such as those related to avocado/raccoon, muffin, parrot, and oil painting style. Linear interpolation generates images not consistent with the prompts, due to issues with non-linear manifolds. CLIP Min. generates images biased towards one of the prompts. Alt. and Step prompt selection methods suffer from artifacts and are not very successful in blending the characteristics of objects corresponding to the individual text prompts - the avocado/raccoon, muffin/dog are not blended well. In the parrot/dog image, the characteristics of the parrot are missing. Alt. generates artifacts in the Times Square/ oil painting image, while Times Square is not characterized well in the image generated by Step. In contrast, the Black-Scholes model adeptly overcomes these limitations, generating realistic images that meticulously balance and preserve the unique characteristics of each individual text prompt. The images are from set 1, set 2, set 3 and set 4 (Refer Section 5.1) respectively.

# 4 BLENDING CONCEPTS USING THE BLACK SCHOLES ALGORITHM

## 4.1 PROBLEM FORMULATION

Consider a set of $N$ text prompts denoted as $P_1, ... P_N$ and $T$ diffusion denoising steps. Note that text-to-image diffusion architectures often include a text encoder that maps the text prompt to a joint text-image space. Starting from random noise and conditioned on the embedding in the text-image space, the neural network generates the final image.

Our objective is to generate an image that aligns with multiple text prompts simultaneously i.e. at the intersection of the various text-image manifolds of the diffusion models. We pose this task as a prompt mixing Patashnik et al. (2023) problem. During each step of diffusion denoising, our aim is to select the most relevant text prompt (or the text prompt with respect to which the image requires further refining) for conditioning the model, ensuring the generation of an optimal image satisfying all individual text prompts.

## 4.2 METHOD

Selecting the most effective text prompt to condition a text-to-image diffusion model can be approached in various ways. Naively switching Kothandaraman et al. (2023a); Patashnik et al. (2023) between prompts is sub-optimal due to the significant human effort involved. Moreover, it fails to address the critical issue of prioritizing the most relevant prompt in an automatic manner. Alternatively, one can compute the CLIP score at each step and move toward the text prompt with the lowest CLIP score. While this approach is reasonable, it overlooks essential factors related to the dynamics Ho et al. (2020) of the diffusion denoising process, which significantly impact the final image generation. Leveraging the Black Scholes model allows us to model these dynamics effectively.

We utilize the CLIP score to quantify the "price" of the generated image, treating it as analogous to a stock. The CLIP score has been widely used to measure text-image alignment, making it a suitable metric for evaluating how well our generated image aligns with each text prompt. Consider diffusion denoising at step $i$ out of total steps $T$. We derive the relevant variables for Black Scholes formula as follows:

1. Underlying stock price $S$: The underlying stock price corresponds to the current value of the asset. Analogously, we define $S$ as the alignment of the generated image with the input text, at the current stage of the diffusion-based generation process. Let $z_t$ represent the predicted latents at timestep $t$. We denote $z_{0,t}$ as the latents of the final predicted image extrapolated from $z_t$. In other words, if the denoising process were to proceed in a straightforward manner following the same direction as computed for $z_t$, the latents of the final predicted image would be $z_{0,t}$. The underlying stock price, denoted as $S$, is determined by the CLIP score with respect to text prompt $P_p$. This CLIP score can be computed using the image decoded from $z_{0,t}$. To maintain values within the range of 0 to 100, we multiply the CLIP score by 100.

2. Strike price $K$: This represents the estimated asset price at expiry. Analogously, we define $K$ as the alignment of the generated image with the input text, at the last step of diffusion denoising, which predicts the final image. We set the strike price as the average CLIP score that the underlying diffusion model achieves when generating an image based on a combination of prompts, using a straightforward approach. This choice is driven by the maximum potential that the diffusion model can realize for that specific set of prompts.

3. Time to expiration $t$: This is the time left before the asset expires. In the case of diffusion model, we define $t$ as the number of steps remaining for the diffusion denoising process to complete, i.e. $t = T - i$.

4. Volatility $\sigma$: This variable refers to the variability in the price of the asset. Analogously, we compute the volatility as the square root of the variance used by the diffusion denoising scheduler at timestep $i$, which is reflective of the volatility in the diffusion process.

5. Risk-free rate $r$: In finance, the risk-free rate is computed as the inflation rate subtracted from the nominal rate. In our formulation w.r.t. diffusion models, the nominal rate can be ignored. The nominal rate in pricing markets i usually set by financial institution based

on various economics factors. In the case of diffusion models, we define it to yield equal proportions of returns over all entire diffusion denoising time-steps, i.e. we assign $r = 1/T$.

During each iteration $i$ of diffusion denoising, we calculate the Black-Scholes score $b_{i,p}$ for each text prompt $P_p$, $p = 1...N$ using the given variables and Equation 1 and Equation 2. Subsequently, during the next step of diffusion denoising, we condition on the text prompt with the lowest Black-Scholes score. This approach ensures that the denoising process prioritizes the text prompt associated with the minimum Black-Scholes score for generating an optimal image consistent with all prompts. Please refer to Algorithm 1 for a step-wise description of the method.

---

**Algorithm 1** Black Scholes prompt mixing in backward diffusion enables the diffusion model generate an optimal image at the intersection of multiple text-image manifolds.

---

1: Initialize latents to random Gaussian noise. $z_T \sim \mathcal{N}(0, I)$; T is the total number of diffusion denoising steps.
2: Use the text encoder $\epsilon$ to compute the CLIP embeddings of a (linguistic) combination of the text prompts $\{P\}$. $e \leftarrow \epsilon_p(\cup\{P\})$.
3: Initialize Black Scholes variables, strike price $K = 0.25 \times 100$, $rater = 1/T$.
4: *// Diffusion denoising - image prediction/generation loop*
5: **for** $t \leftarrow T$ to 0 **do**
6:     $z_{t-1} = z_t - f(z_t, t, e)$; f is the diffusion UNet.
7:     Use $z_{t-1}$ to compute $z_{0,t-1}$.
8:     Variance $\sigma$ at step $t - 1 \leftarrow$ computed using the scheduler of the diffusion model.
9:     **for** $i \leftarrow 1$ to $N$ **do**
10:         Spot price $S \leftarrow$ CLIP score With respect to text prompt $P_i$
11:         Time to expiration $\leftarrow t$
12:         Black Scholes score $b_{t,i}$ with respect to prompt $P_i$, at timestep $t$, $\leftarrow$ use Equation 1,2.
13:     **end for**
14:     $P_{min} \leftarrow$ Text prompt corresponding to $\min\{B_{t,i}\}$, $i = 1...N$
15:     $e \leftarrow \epsilon_p(P_{min})$
16: **end for**

---

## 5 EXPERIMENTS AND RESULTS

**Metrics.** We evaluate performance using the following metrics:

1. **CLIP Score**: We utilize two variants:
   - **CLIP-combined**: This variant assesses the overall alignment with text by comparing the generated image against a combination of individual text prompts.
   - **CLIP-add**: This averages the CLIP scores for the generated image across each individual text prompt, reflecting alignment with each specific concept.

2. **BLIP Score ⊙ DINO**:
   - The **BLIP score** is calculated by comparing the generated image to a combination of individual text prompts, measuring overall text-level alignment.
   - The **DINO score** evaluates the generated image against each individual text prompt, assessing how well the image preserves the characteristics of each concept. This indicates the fidelity of the attributes in the generated image relative to the individual concepts. Thus, while BLIP focuses on overall concept blending, DINO provides insights into the preservation of characteristics for each concept. We aim for high values in both scores, calculating a net score by multiplying the DINO and BLIP scores for each prompt and averaging across all prompts.

3. **KID**: This metric assesses the realism of the generated samples, serving as an indicator of their overall quality.

This structured approach allows us to comprehensively evaluate the performance of generated images in relation to the provided text prompts. To calculate our metrics, we generate five images for each text prompt and incorporate all of them into the metric computations.

**Baselines.** Our study examines several baselines: (i) **Vanilla SD:** Prompt engineering, we use the vanilla stable diffusion model and condition it on a single text prompt effectively describing all individual text prompts, (ii) **Linear Interpolation:** Direct combination of text embeddings, achieved through linear interpolation between text embeddings Kawar et al. (2023). This method equally weights the text embeddings associated with each text prompt. (iii) **Switching between text prompts**. We consider two variations here: (iii-a) **Step:** Following Patashnik et. al. Patashnik et al. (2023), we use one text prompt for the initial 7th to 17th denoising steps and then switch to the other text prompt for the remaining steps, (iii-b) **Alt.:** Following Kothandaraman et. al. Kothandaraman et al. (2023a), we alternate between the two text prompts. (iv) **CLIP Minimum:** Score-based combination, denoted as CLIP-min, where we select the text prompt corresponding to the lowest CLIP score from the previous denoising iteration.

**Hyperparameters.** Based on our experiments for the vanilla combination usig Stable Diffusion 2.1 for the dataset under consideration, where we found that a CLIP score of approximately 0.25 indicates reasonable text-image alignment, we opted for a constant value of 0.25 for the strike price. The ordering of prompts does not matter. This is because, at every step of diffusion denoising, the algorithm chooses the prompt that should be selected by computing the Black Scholes score wrt each prompt, and this process is agnostic to the ordering of the prompts. More details on hyperparameters and the backbone architecture can be found in the appendix.

## 5.1 ANALYSIS AND COMPARISONS

We construct a dataset with 4 types of scenarios to analyze varying complexities, spanning simple text prompts with single object, multiple objects per text prompt, object actions against backgrounds (prompt mixing w.r.t. object) and object performing action against a background/ style (concept blending w.r.t. background and style). More information can be found in the appendix. We show qualitative results in Figure 1 and Figure 2. More qualitative results can be found in the appendix. The quantitative comparisons are in Figure 1. A detailed analysis of our method and benefits over prior art is as follows:

- Vanilla stable diffusion (SD): Vanilla stable diffusion v2.1 can create plausible combinations of the provided text prompts. However, due to hallucination issues, it is constrained by the distributions it learned during training. As a result, it struggles to capture clear characteristics of individual text prompts. For example, in Figure 1, the distinct features of the parrot, cat, dog/penguin, and sunset/penguin are missing. Similarly, in Figure 2, the avocado/raccoon, muffin, parrot, and oil painting style lack distinct characteristics.

- Linear interpolation Kawar et al. (2023): Linear interpolation is ill-suited for generating an image that combines two text prompts. This limitation arises from the non-linear nature of the text-image space. While linear interpolation assumes a simple linear relationship, the actual mapping between textual descriptions and image features is intricate. Furthermore, linear interpolation can only traverse a straight path between two points in the latent space, failing to capture nuanced variations or produce novel features beyond the endpoints.

- Alternating Sampling Kothandaraman et al. (2023a),: Alternating sampling generates features related to both text prompts in the final output image. However, the objects produced suffer from poor definition, unrealistic appearance, and low quality in many instances. Artifacts are prevalent, and several generated images appear implausible, resulting in a high KID score. The underlying issue lies in the algorithm's routine alternation between the two text prompts during diffusion denoising, without adequately emphasizing the most relevant prompt at each step. Despite these limitations, the model achieves a reasonable CLIP Score by broadly aligning with the overall image description.

- Step-wise switching Kothandaraman et al. (2023a): Step-wise switching is similarly ineffective for the same reasons as Alternating sampling.

- CLIP-min: The CLIP-min results exhibit bias toward one of the text prompts in most cases, preventing the model from generating an image that aligns with the intersection of both prompts. This bias arises because the model overlooks the dynamics of diffusion denoising and instead selects the text prompt with the lowest CLIP score at each step, resulting in bias-related issues.

| Method | CLIP-combined ($\uparrow$) | CLIP-add ($\uparrow$) | BLIP $\odot$ DINO ($\uparrow$) | KID ($\downarrow$) | Steps ($\downarrow$) | Time (s) ($\downarrow$) | GPU hrs | Memory (GB) ($\downarrow$) |
|---|---|---|---|---|---|---|---|---|
| Linear Int. Kawar et al. (2023) | 0.2885 | 0.2778 | 0.2588 | 0.02851 | 50 | 6.5 | 0.001805 | 7.1 |
| Alt. Samp. Kothandaraman et al. (2023a) | 0.3445 | 0.3098 | 0.3894 | 0.01786 | 100 | 14 | 0.00389 | 7.7 |
| CLIP Min. | 0.3195 | 0.2955 | 0.3107 | 0.00866 | 100 | 14 | 0.00389 | 7.7 |
| Step Patashnik et al. (2023) | 0.3220 | 0.2997 | 0.3390 | 0.01709 | 100 | 14 | 0.00389 | 7.7 |
| Black Scholes | **0.3469** | **0.3112** | **0.3912** | **0.01531** | **100** | **14** | **0.00389** | **7.7** |

Table 1: We evaluate various properties using CLIP Scores, BLIP Scores, DINO, and KID. These metrics help us assess overall text alignment with the combined text prompts, the preservation of attributes related to individual concepts, and the quality of the generated images. The Black Scholes algorithm for prompt mixing in diffusion models achieves superior results, compared to other prompt-mixing techniques, as also evidenced by the qualitative results.

- Black Scholes algorithm: Our approach based on the Black-Scholes algorithm effectively generates realistic images that align with the intersection of the two text prompts. These images exhibit minimal unrealistic artifacts. Additionally, our algorithm successfully preserves individual characteristics corresponding to each text prompt. By modeling the dynamics of diffusion denoising, the algorithm strategically selects the optimal text prompt at each step, achieving a balanced synthesis of both prompts. Notably, our model attains superior quantitative results w.r.t metrics.

In summary, the Black-Scholes model outperforms all previous methods for prompt mixing by generating realistic images that meticulously preserve and balance the characteristics of each individual text prompt.

**Computational complexity**   We report the computational requirements for different methods in Table 1. All values are computed using one NVIDIA A5000 GPU, on the 2 prompts blending scenario. Our algorithm incurs minimal computational overhead over other methods, primarily arising from the computation of the CLIP score of the generated image w.r.t. all individual text prompts at every step of the diffusion denoising based generation process.

## 6    CONCLUSIONS, LIMITATIONS, AND FUTURE WORK

This paper introduces a new method for prompt mixing, inspired by a financial probabilistic model. By comparing the Black-Scholes algorithm to diffusion models, we develop an algorithm that generates images based on multiple text prompts, demonstrating significant qualitative and quantitative benefits. While our method leverages diffusion models, it may not be applicable to non-Gaussian Bansal et al. (2024) diffusion models or one-step diffusion models Yin et al. (2023), as demonstrated in recent research, and is an interesting direction for future work. Moreover, future work on integrating our approach with recent advancements in attention guidance, layout modeling, and related areas could extend its applicability to critical downstream tasks such as image editing Kawar et al. (2023); Yang et al. (2023a); Avrahami et al. (2022), compositionality Liu et al. (2022); Agarwal et al. (2023), handling complex prompts Lian et al. (2023), text-based view synthesis Kothandaraman et al. (2023a;b) and personalized image generation Ruiz et al. (2023); Kumari et al. (2023). While recognizing the success of recent text-to-image models like Dalle-3, Stable Diffusion 3, Firefly 3, and Imagen 3 in generating images from diverse and complex text prompts, we introduce a data-efficient approach to prompt mixing. Our method offers advantages over previous techniques for prompt-mixing, and future extension could explore integrating our model's benefits into state-of-the-art text-to-image architectures for data efficient prompt mixing and downstream applications.

**Societal impact.**   While our method offers valuable tools for generative AI-based content creation, its potential misuse underscores the need for research in watermarking and deepfake detection to mitigate risks.

**Acknowledgements**   will be inserted in the final version of the paper.

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

## A.1. EXPERIMENTAL SETTINGS

We consider the following experimental settings:

**Set 1:** Simple text prompts with single objects. In this experiment, we validate our approach using straightforward scenarios, we use two text prompts, each describing one class. Specifically, we consider 17 classes: ['a rock', 'a coffee mug', 'a cute dog', 'a pink cat', 'a teddy bear', 'a robot', 'an alien', 'an avocado', 'a cute raccoon', 'a corgi dog', 'a parrot', 'a car', 'a squirrel', 'a cute rabbit', 'pizza', 'muffin', 'icecream'], and construct $17c_2 = 136$ text prompts.

**Set 2:** Multiple objects per text prompt. To add complexity, blend multiple objects, in the presence of additional objects in the scene. The goal of this experiment is to assess the models' capabilities in blending the right objects in the scene, and preserving the characteristics of the other objects. We work with three classes of objects: ['a basket', 'a teapot'], ['apples', 'bananas', 'a cute cat', 'a cute dog', 'muffins'], ['a table', 'a carpet', 'a bed'] and construct combinations using the first and second set and the second and third set. Additionally, we include the data point "a cat in a bathtub" and "a corgi dog in a bathtub," resulting in a total of 71 prompts.

**Set 3:** Object actions against backgrounds (prompt mixing w.r.t. object). We investigate the model's ability to morph objects while considering object-level action information and scene or background context. We consider the following backgrounds, ['walking in Times Square', 'skiing in Times Square', 'walking in a beautiful garden', 'surfing on the beach', 'eating watermelon on the beach', 'sitting on a sofa on the beach', 'watching northern lights', 'watching sunset at a beach', 'admiring the opera house in Sydney', 'sleeping in a cozy bedroom', 'admiring a beautiful waterfall in a forest', 'walking in a cherry blossom garden', 'walking in a colorful autumn forest', 'flying in the sky at sunset']. The objects are ['a kangaroo', 'a cute cat', 'a corgi dog', 'a parrot', 'a teddy bear', 'a penguin']. In total, we create 210 prompts.

**Set 4:** Object performing action against a background (prompt mixing w.r.t. background). In this experiment, we explore how the model blends global information and backgrounds while considering objects in various scenarios. We consider the objects ['a kangaroo', 'a cute cat', 'a corgi dog', 'a parrot', 'a teddy bear', 'a penguin']. The backgrounds and style prompts are ['walking in Times Square', 'sunset', 'van gogh style', 'oil painting', 'a garden with tulips', 'northern lights']. Overall, we have 90 prompts in this experiment.

### A.1.1. HYPERPARAMETERS:

We use the Stable Diffusion 2.1 backbone in all our experiments. We use standard diffusion parameters: image size - (512, 512); classifier-free guidance scale - 7.5; DDIMScheduler with betaStart set to 0.00085, betaEnd set to 0.012, betaSchedule set to "scaledLinear", clipSample set to False and setAlphaToOne set to False. We use 100 inference steps in all our experiments.

For the set of prompts in our dataset under consideration, the average CLIP score for the combination of prompts in the vanilla case was 0.25. Hence, we set the value of strike price $K$ at 0.25.

**Backbone architecture.** We use the Stable Diffusion 2.1 backbone model Rombach et al. (2022) in all our experiments. There is no training involved, we use the pretrained model to directly perform inference. Our experiments are run on one NVIDIA A5000 GPU with 24 GB memory, and takes 14 seconds to generate each image using the Black Scholes model, the diffusion denoising is performed for 100 steps. For comparison, the vanilla stable diffusion model takes 13 seconds for 100 steps on the same GPU.

## A.2. ASSUMPTIONS MADE BY THE BLACK SCHOLES ALGORITHM

The Black Scholes model makes five assumptions: (i) No dividends are paid out during the life of an option, (ii) Market movements are somewhat random, (iii) There are no transaction costs in buying the asset, (iv) The volatility and risk free rate of the underlying asset are known and constant, (v)

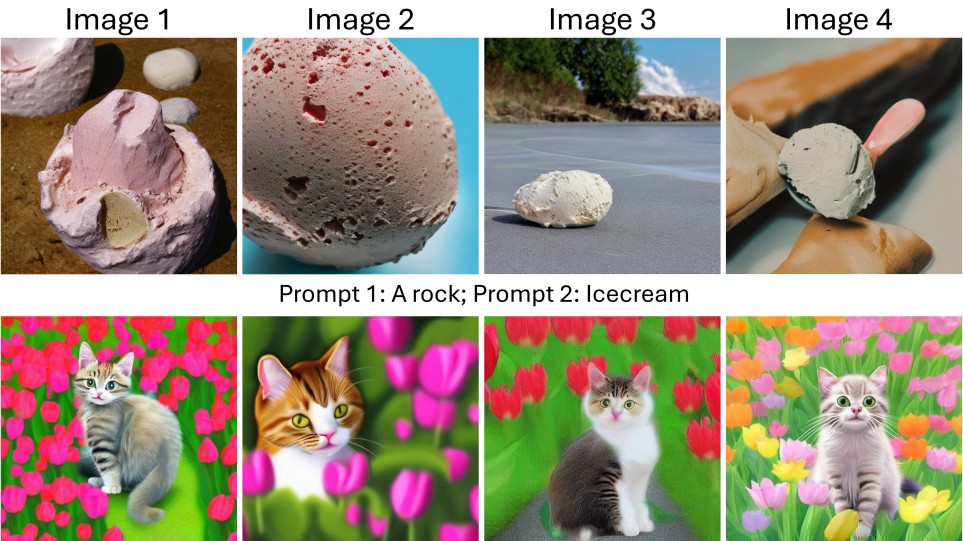

Prompt 1: A rock; Prompt 2: Icecream

Prompt 1: A cute cat walking in a tulip garden; Prompt 2: a cute cat in oil painting style

Figure 3: Image variations from our method using the Black Scholes model for a single prompt, starting from different random Gaussian noise initializations.

The returns of the underlying asset are normally distributed. The relation between the assumptions made by the Black Scholes algorithm and diffusion models is as follows:

- No Dividends (Assumption i): This assumption is specific to financial options and does not directly apply to diffusion models. In the context of diffusion models, we can consider that the "output" (the generated image) does not have any intermediate rewards or returns, similar to how options do not yield dividends.

- Random Market Movements (Assumption ii): This assumption aligns well with the inherent randomness in the diffusion process. Just as stock prices are subject to unpredictable fluctuations, the generation of images in diffusion models involves stochastic processes where the final output is influenced by random noise and iterative refinement.

- No Transaction Costs (Assumption iii): Similar to the first assumption, transaction costs are not relevant in the context of diffusion models. Instead, we can think of the computational resources and time required for generating images as analogous to transaction costs, but they do not affect the model's fundamental operation.

- Constant Volatility and Risk-Free Rate (Assumption iv): In diffusion models, while volatility is not explicitly defined as in financial contexts, the concept of stability in the generation process can be likened to having a consistent framework for how noise is added and refined. The "risk-free rate" does not have a direct counterpart, but the idea of a stable environment for generating images can be considered.

- Normally Distributed Returns (Assumption v): This assumption can be related to the distribution of pixel values or features in the generated images. In many diffusion models, the outputs are conditioned on a distribution that can be approximated as normal, especially when considering the latent space representations.

In summary, while some assumptions of the Black-Scholes model are not directly applicable to diffusion models, others, particularly the randomness and distribution aspects, provide a meaningful connection, which leads to our algorithm on how concepts from finance can inform generative modeling techniques!

Vanilla SD    Lin. Interp.    Alt.    Step    CLIP Min.    Black Scholes

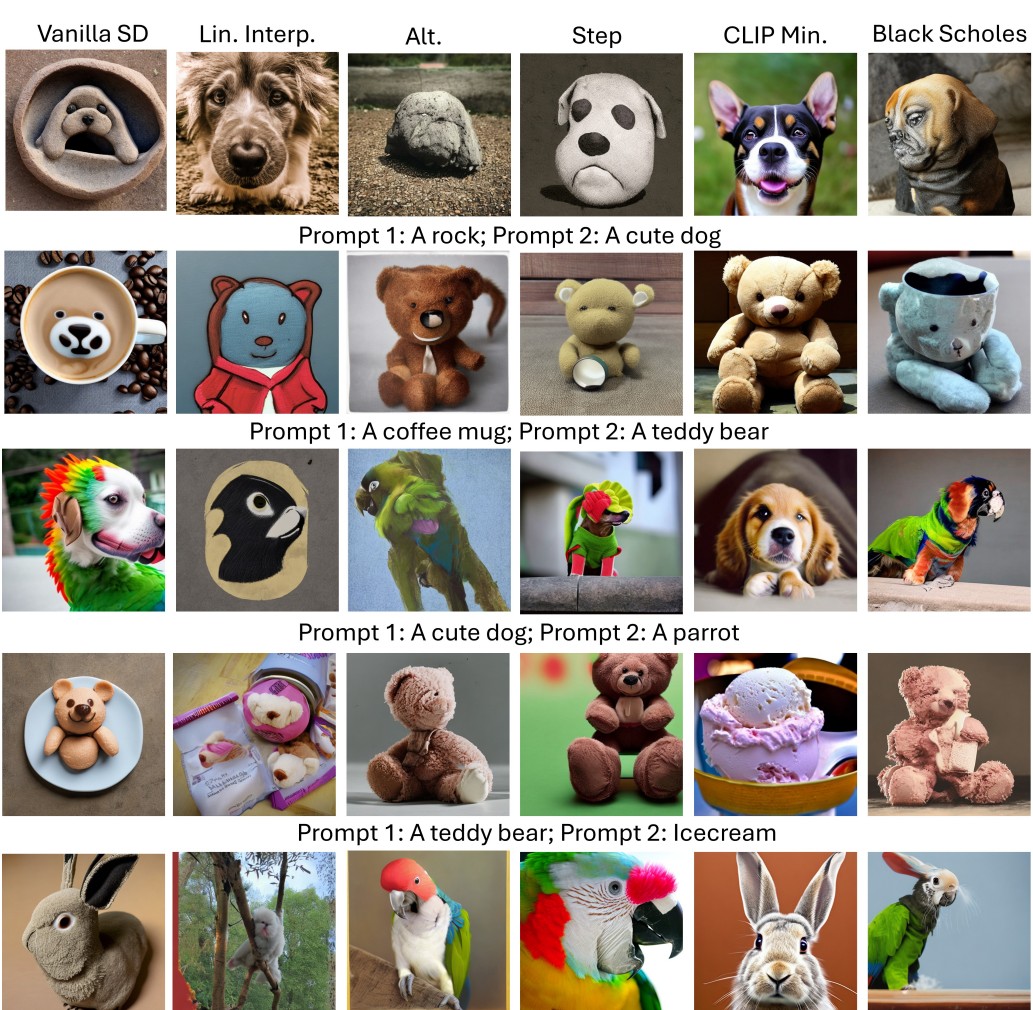

Prompt 1: A rock; Prompt 2: A cute dog

Prompt 1: A coffee mug; Prompt 2: A teddy bear

Prompt 1: A cute dog; Prompt 2: A parrot

Prompt 1: A teddy bear; Prompt 2: Icecream

Prompt 1: A parrot; Prompt 2: A cute rabbit

Figure 4: Vanilla stable diffusion (SD) is able to generate images satisfying both text prompts in many cases, however it misses out on the fine-grained characteristics of the individual text prompts. For instance, it misses out on the characteristics of the dog, parrot and ice-cream. Linear interpolation generates images that are not consistent with the text prompts. CLIP Min. generates images biased towards one of the text promots. Alternating sampling and Step wise inference strategies generate images with a lot of artifacts, and the generated images are perceptually implausible in many cases. For instance, in order, the issues are: missing characteristics of dog, missing characteristics of coffee mug and artifacts in teddy bear, artifacts and missing characteristics of parrot, missing characteristics of teddy bear and artifacts in teddy bear, missing characteristics of rabbit. Our Black Scholes method is able to generate images that capture the fine-grained characteristics of objects corresponding to both text prompts, and the generated images look realistic with minimal artifacts. The text prompts are from set 1.

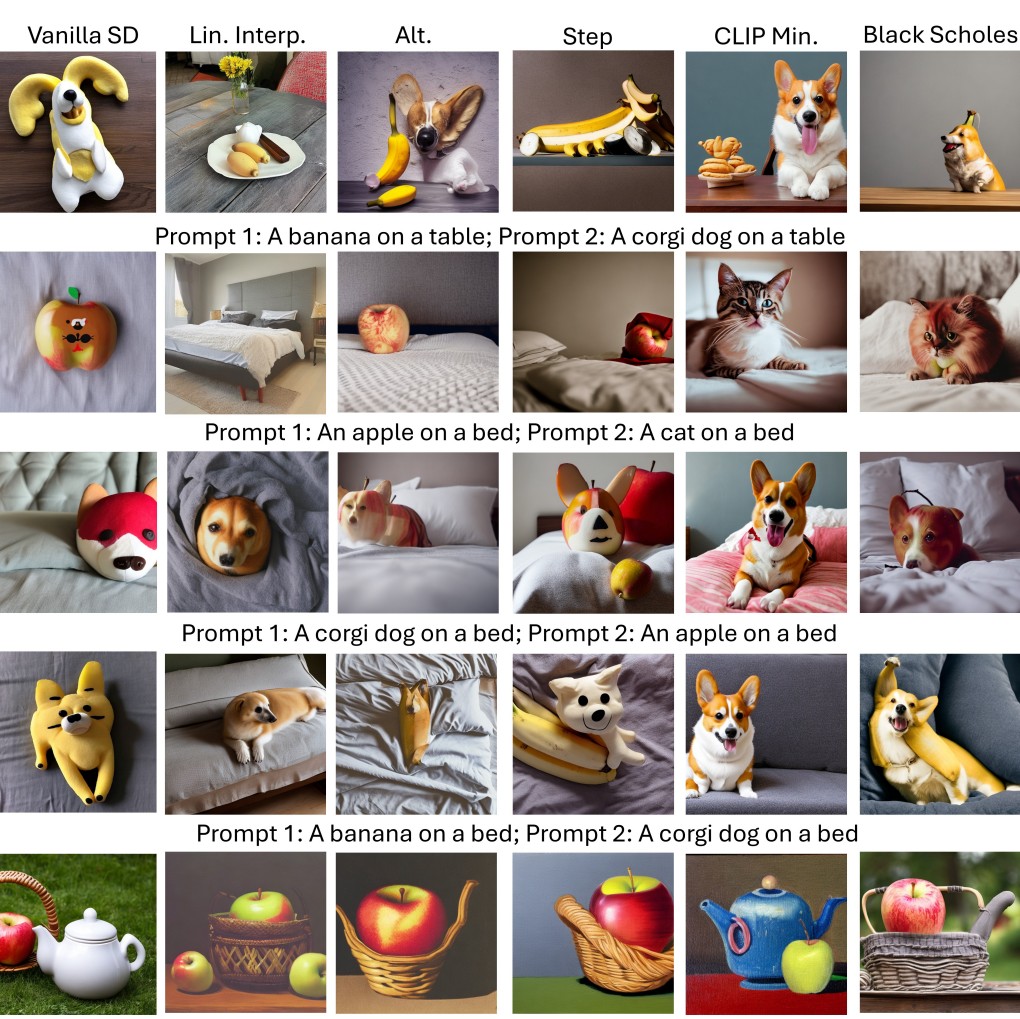

Prompt 1: A banana on a table; Prompt 2: A corgi dog on a table

Prompt 1: An apple on a bed; Prompt 2: A cat on a bed

Prompt 1: A corgi dog on a bed; Prompt 2: An apple on a bed

Prompt 1: A banana on a bed; Prompt 2: A corgi dog on a bed

Prompt 1: A teapot with an apple; Prompt 2: A basket with an apple

Figure 5: Vanilla stable diffusion (SD) is able to generate images satisfying both text prompts in many cases, however it misses out on the fine-grained characteristics of the individual text prompts. For instance, it misses out on the characteristics of the dog, cat, corgi dog and basket/ teapot mixing. Linear interpolation generates images that are not consistent with the text prompts. CLIP Min. is biased towards one of the text prompts. Alternating sampling and Step wise inference strategies generate images with a lot of artifacts, and the generated images are perceptually implausible in many cases. For instance, in order, the issues are: artifacts in dog/banana and missing characteristics of dog, missing characteristics of cat, the corgi dog in the dog/apple image does not describe the dog as well as our Black Scholes method does, the dog is not well described in the fourth example, the teapot is not well described in the final image. Our method is able to generate images that capture the fine-grained characteristics of objects corresponding to both text prompts, and the generated images look realistic with minimal artifacts. The text prompts are from set 2.

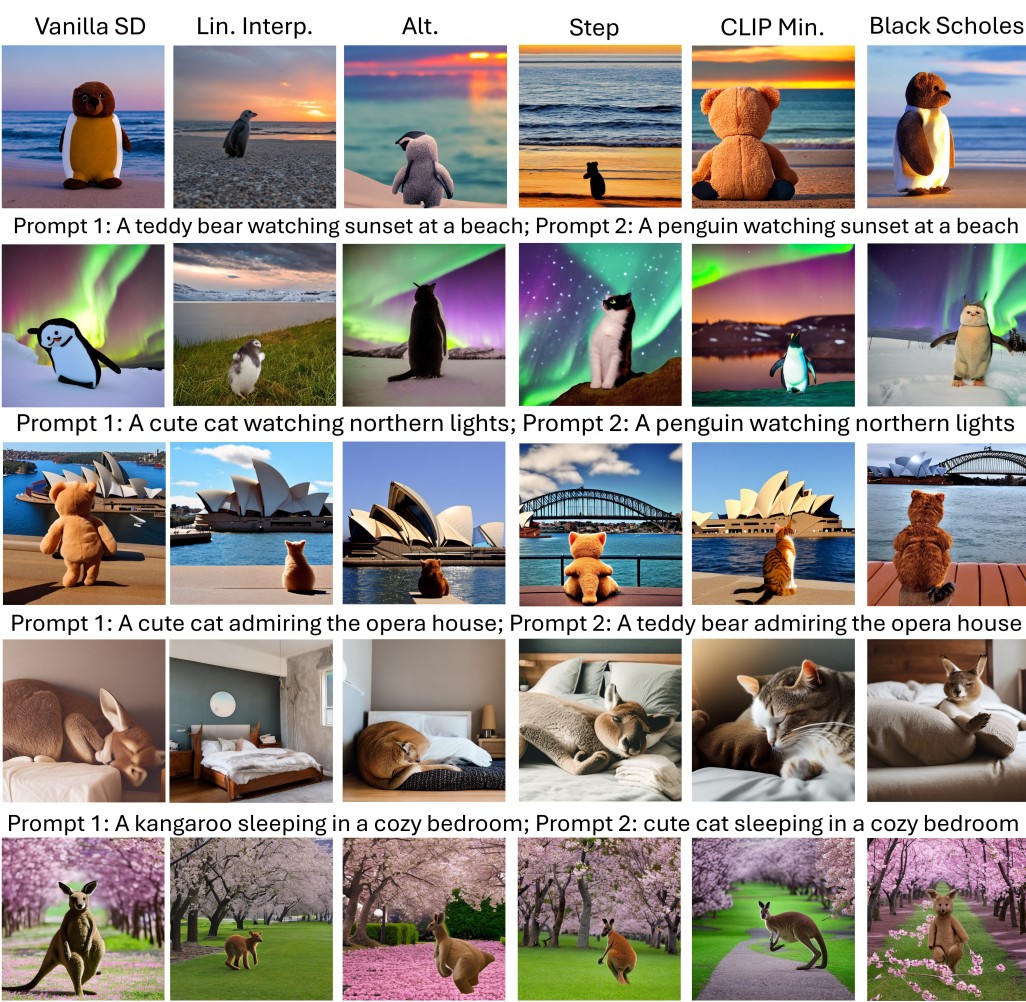

Figure 6: Vanilla stable diffusion (SD) is able to generate images satisfying both text prompts in many cases, however it misses out on the fine-grained characteristics of the individual text prompts. For instance, in the first and fifth image, the characteristics of the teddy bear are missing, the second, third and fourth image do not describe the cat well, Linear interpolation generates images that are not consistent with the text prompts. CLIP Min is again biased towards one of the text prompts. Alternating sampling and Step wise inference strategies generate images with a lot of artifacts, and the generated images are perceptually implausible in many cases. Specifically, the first image does not describe the subjects well, the second and third images are reasonably generated but our Black Scholes method generates a clearer image of the penguin blended cat, and teddy bear blended cat, the results of Alt. for the fourth case has artifacts and Step does not describe the cat well, the teddy bear is not well described in the fifth image. Our Black Scholes method is able to generate images that capture the fine-grained characteristics of objects corresponding to both text prompts, and the generated images look realistic with minimal artifacts. The text prompts are from set 3.

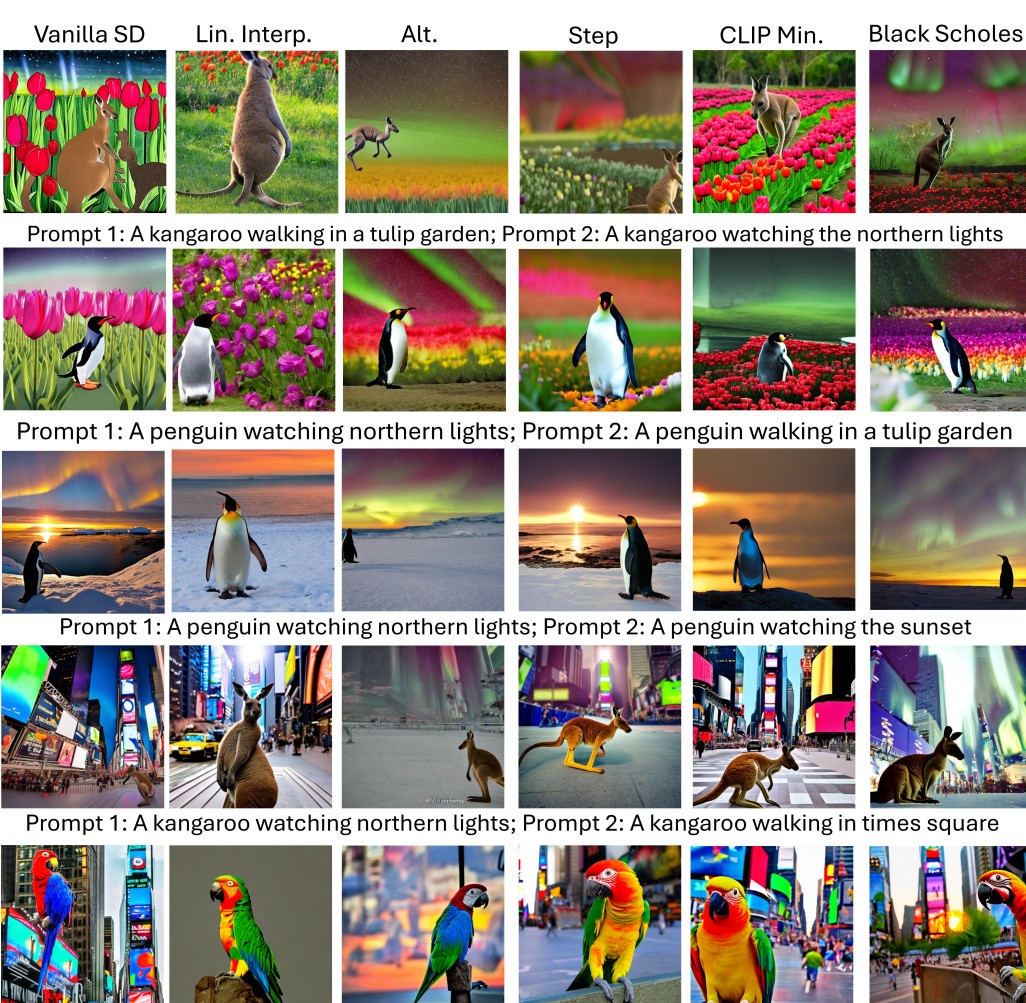

| Vanilla SD | Lin. Interp. | Alt. | Step | CLIP Min. | Black Scholes |
|---|---|---|---|---|---|

Prompt 1: A kangaroo walking in a tulip garden; Prompt 2: A kangaroo watching the northern lights

Prompt 1: A penguin watching northern lights; Prompt 2: A penguin walking in a tulip garden

Prompt 1: A penguin watching northern lights; Prompt 2: A penguin watching the sunset

Prompt 1: A kangaroo watching northern lights; Prompt 2: A kangaroo walking in times square

Prompt 1: A parrot watching the sunset; Prompt 2: A parrot walking in times square

Figure 7: Vanilla stable diffusion (SD) is able to generate images satisfying both text prompts in many cases, however it misses out on the fine-grained characteristics of the individual text prompts. The first two images are cartoon-ish, and Northern lights is not clearly depicted. The third image also misses a clear description of Northern lights. In the fourth image, some of the billboards in Times Square are green, but there is no sign of Northern lights. In the fifth image, the sunset is missing. Linear interpolation generates images that are not consistent with the text prompts. CLIP Min results are biased towards one of the prompts for all images, except the second one. Alternating sampling and Step wise inference strategies generate images with a lot of artifacts, and the generated images are perceptually implausible in many cases. In the first image, the kangaroo is not generated well (almost flying in Alt's result, and the characteristics of the kangaroo are not well generated by Step). In the second image, the characteristics of the tulip garden are not well represented. In the third image, notice how our model generates the characteristics of the sunset and Northern lights better, while ensuring that they are blended well to form a realistic image. In the fourth image, Alt and Step miss out on the characteristics of Times Square and Northern lights respectively. In the fifth image, Alt and Step miss out on the characteristics of Times Square and the sunset respectively. Our Black Scholes method is able to generate images that capture the fine-grained characteristics of objects corresponding to both text prompts, and the generated images look realistic with minimal artifacts. The text prompts are from set 4.

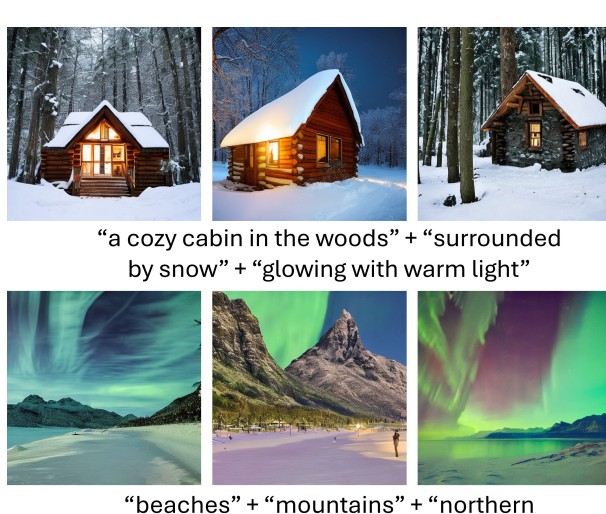

"a cozy cabin in the woods" + "surrounded by snow" + "glowing with warm light"

"beaches" + "mountains" + "northern lights"

Figure 8: Our method can be extended to concept blending involving more than 2 prompts (3 in this case), as shown above.

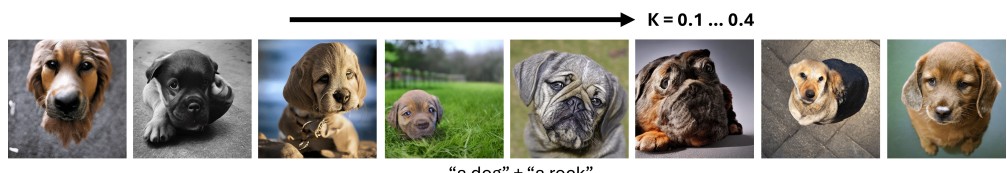

K = 0.1 ... 0.4

"a dog" + "a rock"

Figure 9: Ablation experiment on the strike price $K$ reveals that the results are optimal at values of $K$ close to the CLIP score of the image generated in the vanilla case using a combination of the constituent text prompts.

