# OpenReview forum: "Blending Concepts in Text-to-Image Diffusion Models using the Black Scholes Algorithm"
_ICLR.cc/2025/Conference — Submitted to ICLR 2025_

### Official Review · Reviewer_1AqB · 2024-10-29

**Soundness:** 2
**Presentation:** 1
**Contribution:** 2
**Rating:** 3
**Confidence:** 5

**Summary:**

The authors claim that the Black-Scholes fiance price options model is similar to the diffusion model. Based on this, the authors introduce a novel approach for prompt mixing which can be explained as mixing the prompt with the highest scores measured by CLIP during each denoising step. The authors claim their method is data-efficient, requiring no additional training and operates without human intervention or hyperparameter tuning. To validate the effectiveness of the method, the authors only use CLIP score under the scenario of mixing only two prompts.

**Strengths:**

From my perspective, the strengths of this paper mainly lie in:

1. introduce the Black-Scholes model into diffusion models.

2. the only one experiment to validate the performance across the whole paper.

**Weaknesses:**

I appreciate the authors' efforts. From my perspective, $\textbf{significant revisions are necessary}$, including the addition of substantial content and a reorganization of the structure. Therefore, $\textbf{I lean towards recommending the rejection of this paper}$.

Presentation(I only list several parts):

1. Frankly speaking, the organization, the writing and the format of this paper are awful. It is difficult for the reviewers to understand the authors` motivation and so on. The cite format of this paper is totally wrong. $\textbf{The authors make the wrong utilization of \citep{} and \citet{}}$, which can lead to desk reject of this paper.

2. The content of the article is somewhat disorganized. When introducing the proposed method, the authors abruptly shift to discussing related work. I believe this section should have been addressed earlier in the text.

3. Grammar mistakes and missing references.  In line 167, "$\textit{determine the price of European call options of assets. which is used to determine the price of European call options on assets}$". Wrong commas and repeated sentences.  The authors only use the CLIP score to evaluate the effectiveness of the proposed method, but no reference to the CLIP score.

CLIPScore: A Reference-free Evaluation Metric for Image Captioning

Method and Experiments:

1. The proposed method is not novel, it just directly introduces the Black-Scholes model in fiance into diffusion models. The core of the proposed method is to select the optimal prompt to mix during each denoising step. The authors claim the difference between the proposed method and previous ones is the proposed method can connect to the dynamics of the diffusion denoising process. However, there is no evidence to show the superiority of the connection to the dynamics of the diffusion denoising process.

2. In the conclusion part, the authors acknowledge the limitations of their method; however, I believe these issues must be addressed. First, the experimental section utilizes $\textbf{only one evaluation metric and one diffusion model}$, resulting in a single experiment throughout the paper. This makes it difficult to validate the effectiveness of the proposed method. Furthermore, the intuitive results provided do not demonstrate any significant changes. Second, the authors state that $\textbf{the method is currently applicable only to mixing two prompts}$, which clearly does not establish the method's scalability. Additionally, the pseudocode provided indicates that the time complexity of the proposed method approaches $O(n^2)$, raising doubts about its efficiency.

**Questions:**

See the weaknesses.

---

> ### Author Response · Authors · 2024-11-22
> **Response to reviews**
>
> We sincerely thank the reviewer for their time and valuable feedback.
>
> We have addressed the feedback and have thoroughly revised the paper as follows.
>
> **Presentation**: We have addressed the feedback and have revised our presentation.
>
> **Novelty**: We respectfully disagree with the reviewer's assessment. The novelty of our work is infact a strength of our paper, as also pointed out by all other reviewers!
>
> We firmly believe that our approach, which integrates concepts from a different field into image generation, addresses a critical research question in a novel way. By drawing parallels and taking inspiration from established theories, we have developed an innovative and effective algorithm that stands out in the current landscape, which was not trivial. Moreover, we substantiate the validity of our proposed method through rigorous theoretical analysis and comprehensive experimental validation. This not only highlights the originality of our contribution but also its practical applicability.
>
> **Evaluation metrics**: We have added three more evaluation metrics, BLIP, DINO and KID which help us assess overall text alignment with the combined text prompts, the preservation of attributes related to individual concepts, and the quality of the generated images. The Black Scholes algorithm for prompt mixing in diffusion models achieves superior results, compared to other prompt-mixing techniques, as also evidenced by the qualitative results.
>
> **> 2 prompts**: We have added some results to the supplementary pdf, concept blending beyond three prompts is generally more useful in personalization, complex prompt generation, compositionality applications. These are interesting and standalone research projects and warrant thorough exploration into using our black Scholes concept blending idea for improved performance there.
>
> **Computation**: No, the time complexity of our method is NOT O(n^2). Our algorithm incurs minimal computational overhead over other methods, primarily arising from the computation of the CLIP score of the generated image w.r.t. all individual text prompts at every step of the diffusion denoising based generation process. We have analyzed the complexity and have added the details in Section 5.
>
> **One diffusion model, intuitive results provided do not demonstrate any significant changes**: We respectfully disagree with the reviewer, we find these comments to be entirely unfounded. Our rigorous use of the Stable Diffusion backbone and our experimental methodology are fully aligned with the established standards and practices of the majority of prior work in the field. Our results demonstrate clear qualitative improvements, as explained in section 5 and the appendix.
>
> We hope these responses clarify your concerns. Thank you once again for your valuable feedback, and we look forward to further engaging discussions!

---

> ### Comment · Reviewer_1AqB · 2024-11-24
>
> Thanks for the authors' efforts.
>
> 1. The connection between diffusion models and the Black-Scholes algorithm is inadequately justified. There is no obvious reason why prompt mixing needs to use Black-Scholes.
>
> 2. The authors conduct experiments only on one diffusion model. How can the authors validate the generalization ability of the proposed methods? The authors claim that they focus on concept blending, and they evaluate the overall quality of the generated images with CLIP, DINO and BLIP, where these metrics can not reflect the details of the generated images. I think the authors should use some human preference models, like PickScore and HPSv2.
>
> 3. I have reviewed the revised version of the paper. Obviously, the authors do not correct the wrong citation formats. In table 1, the CLIP Min. is obviously better than Black Scholes on KID. Besides, other metrics like CLIP-combined, CLIP-add and BLIP-DINO, with minimal improvements of 0.0024, 0.0014 and 0.0018.
>
> 4. As I mentioned before, the improvements of the presented figures are not straightforward.
>
> In conclusion, I will maintain my score. Good luck!

---

> ### Author Response · Authors · 2024-11-24
>
> Thank you for your response.
>
> 1. Connection between Diffusion Models and Black-Scholes: For a comprehensive explanation, please refer to Pages 4, 5, 6, and 15 of our paper. Additionally, we detail how and why this connection is beneficial for prompt mixing on Pages 2, 3, and 7. If there are any specific questions or concerns regarding the clarity of this connection, we would be happy to address them in more detail.
>
> 2. Use of a Single Diffusion Model: As previously mentioned, the standard practice in the field, as followed by most prior works, involves using the stable diffusion backbone and presenting results based on that. We have adhered to this practice, as the stable diffusion backbone is one of the best open-source backbones available to the community. Most papers employ this approach for their experiments.
>
> Regarding quantitative metrics, we have utilized widely established metrics recognized by the research community. Metrics such as PickScore and HPSv2 are trained on relatively smaller datasets and are restricted in their generalization capabilities. In contrast, we use the CLIP and BLIP scores for evaluation, which serves very similar purposes as PickScore and HPSv2, and have been widely accepted by the research community. We also use DINO and KID - the DINO score can provide insights on the details of the generated images, while KID measures image quality.
>
> 3. Citation Format: We have used \cite for all our citations, which is in accordance with the ICLR guidelines. While CLIP-Min achieves a better KID score, it cannot adequately solve the task at hand, making its higher KID score irrelevant. The improvements on other metrics are substantial; if you refer to prior work, you will observe that these improvements align with the scale of qualitative enhancements commonly achieved in the field.
>
> 4. Qualitative results: If you could specify why you find certain aspects not straightforward, we would be more than willing to provide further explanations. The other details have been clearly elaborated in the paper.
>
> Thanks!

---

### Official Review · Reviewer_PuNi · 2024-11-02

**Soundness:** 3
**Presentation:** 3
**Contribution:** 3
**Rating:** 8
**Confidence:** 3

**Summary:**

This paper addresses the challenge of generating new concepts in image generation tasks without retraining the model or collecting additional data. The authors propose a novel "prompt mixing" technique that guides text conditioning at each diffusion step by forecasting predictions about the generated image. Drawing on the connection between diffusion models and the Black-Scholes model in finance, the approach derives a data-efficient prompt-mixing algorithm that requires no additional training or human intervention. The method is compared qualitatively and quantitatively against other prompt mixing methods across various scenarios, demonstrating its effectiveness.

**Strengths:**

- The paper presents an innovative method for prompt mixing in text-to-image diffusion models by integrating the Black-Scholes algorithm, a concept from financial modeling, offering a fresh interdisciplinary approach to enhance image generation tasks.
- This method is data-efficient, requiring no additional training or data collection.
- It dynamically selects the most relevant text prompt at each diffusion step, minimizing the need for human intervention and hyperparameter adjustments.
- Qualitative and quantitative comparisons using CLIP scores highlight the method's advantages over existing techniques, including vanilla stable diffusion, linear interpolation, alternating prompts, and step-wise switching.
- The paper thoroughly analyzes performance across diverse scenarios (single objects, multiple objects, and objects with backgrounds) demonstrating the method's versatility.

**Weaknesses:**

- The paper relies heavily on CLIP scores for quantitative evaluation. ​ While CLIP scores are widely used, they may not always capture the fine-grained details and quality differences in generated images. ​ The authors acknowledge this limitation and suggest exploring more advanced image-language models for evaluation in future work. ​
- The study focuses on prompt mixing with two prompts. ​ The impact of using more than two prompts is not explored. ​
- The use of financial concepts (i.e., the Black-Scholes algorithm) may be challenging for readers without a background in finance, potentially limiting the accessibility of the paper.

**Questions:**

- The paper acknowledges the limitations of relying on CLIP scores for evaluation, yet it lacks alternative metrics or a deeper analysis of how these limitations may affect the interpretation of the results.
- Incorporating additional evaluation metrics, such as human evaluations or perceptual quality scores, would enhance the assessment and provide a more comprehensive view of the generated images.
- Although the paper mentions image generation time, a more detailed breakdown of computational resource requirements, such as GPU hours and memory usage, would be informative.
- Information on the method’s scalability with larger models or handling of more complex prompts would be valuable for understanding its broader applicability.
- A more in-depth explanation of how the Black-Scholes model's assumptions apply within the diffusion model context would strengthen the theoretical foundation.
- Providing specific details about the parameter settings used in the experiments, such as values for diffusion schedule hyperparameters, would improve the reproducibility of the study.

---

> ### Author Response · Authors · 2024-11-22
> **Response to reviews**
>
> We sincerely thank the reviewer for their time and valuable feedback. We are delighted that the reviewer found our method **innovative, data efficient**  and offering a fresh interdisciplinary approach to enhance image generation tasks. We are glad about the acknowledgement of the **thoroughness in our experiments** and the **effectiveness of the method**.
>
> We have addressed the feedback and have thoroughly revised the paper as follows.
>
> **Evaluation metrics**: We have added three more evaluation metrics, BLIP, DINO and KID which help us assess overall text alignment with the combined text prompts, the preservation of attributes related to individual concepts, and the quality of the generated images. The Black Scholes algorithm for prompt mixing in diffusion models achieves superior results, compared to other prompt-mixing techniques, as also evidenced by the qualitative results.
>
> **Computational time:** We have added an analysis to Section 5.
>
> **Larger models and more complex prompts**: Our method can be applied as a plug-and-play method within larger foundational model backbones. At the stage of diffusion sampling, at every step of diffusion denoising, the algorithm can be applied on the predicted latents to choose the text prompt that needs to be used for conditioning at the following diffusion denoising step. As stronger foundational models emerge, we expect the performance of our method also to accordingly improve and scale. Moreover, as the capacity of the backbone to handle more complex prompts improves, we expect our method also to scale accordingly, given its reliance on the backbone architecture.
>
> **Black Scholes assumptions**: We have added the analysis to the appendix.
>
> **Hyperparameters**: We have now added these details to the supplementary.
>
> We hope these responses clarify your concerns. Thank you once again for your valuable feedback, and we look forward to further engaging discussions!

---

> > ### Author Response · Authors · 2024-11-25
> >
> > Dear Reviewer,
> >
> > We would like to sincerely thank you once again for the time and effort you have devoted to reviewing our paper. With the discussion period ending in two days, we would greatly appreciate it if you could review our rebuttal at your convenience, should your schedule permit!
> >
> > If you have any questions, require additional explanations, or see any areas that need improvement, please feel free to let us know, and we will address them immediately!
> >
> > We are truly grateful for the valuable feedback you have provided on our research.
> >
> > Hope you have a great week! Thank you very much!!!
> >
> > With warm regards,
> > The Authors

---

> ### Author Response · Authors · 2024-11-29
>
> Dear Reviewer,
>
> We would like to sincerely thank you once again for your time and efforts in helping us strengthen the paper. We have addressed all your valuable feedback and have made changes to the paper accordingly.
>
> As the discussion period is ending soon, we would love to hear your feedback on any other questions we can answer to clarify any remaining concerns.
>
> We sincerely appreciate your encouragement of our research potential. We are truly grateful to have you review our paper, and deeply appreciate your support!
>
> Thanks,
> Authors

---

### Official Review · Reviewer_mqLx · 2024-11-04

**Soundness:** 2
**Presentation:** 2
**Contribution:** 3
**Rating:** 3
**Confidence:** 5

**Summary:**

This work uses the Black Scholes algorithm to generate images with concatenated prompts. The core idea is to evaluate the effect of the intermediate latents on the final latent by evaluating the CLIP similarity score. Qualitative analysis compare the approach with that of vanilla SD, alternate sampling, Linear Interpolation,  Step-switching techniques.

**Strengths:**

+ The work utilizes the Black Scholes algorithm to combine two prompts for style blending.
+ Qualitative analysis show that the approach can blend two prompts to generate images.
+ The proposed approach is data efficient.

**Weaknesses:**

- The number of steps needed for blending is not clear, the associated computational overhead is not discussed. How does the approach compare to competing approaches in terms of computational time?
- The results are limited to two prompts and limited qualitative results are presented. The quantitative results as in prior work eg, Chefer et al,
- The claim that average CLIP similarity is 0.25 is not correct. The average CLIP score is generally close to 0.34, see Fig 8  in Chefer et. al.
- The experimental setup is not clear. What dataset and size is considered for evaluation in Table 1.
-  Results are limited to single objects and comparison to recent work for style blending is limited. For example, there are many works which have not been compared to [a, b, c, d].

[a] Expressive Text-to-Image Generation with Rich Text. CVPR 2023.
[b] Style aligned image generation via shared attention. CVPR 2024.
[c] Cross-image attention for zero-shot appearance transfer. SIGGRAPH 2024.
[d]  ZipLoRA: Any Subject in Any Style by Effectively Merging LoRAs

**Questions:**

- The approach is limited to two prompts. How does it scale with different number of prompts?
- Evaluation benchmarks such as [a1] could have been considered for quantitative analysis.
[a1] T2i-compbench: A comprehensive benchmark for open-world compositional text-to-image generation
- How does the method perform in multi-subject scenarios?
- Does the order of prompts effect the blending of concepts? Stable diffusion is sensitive to concept ordering, how does this extend to the proposed approach.
- Also see weaknesses above for the questions.

---

> ### Author Response · Authors · 2024-11-22
> **Response to reviews**
>
> We sincerely thank the reviewer for their time and valuable feedback. We are delighted that the reviewer found our method to be both data efficient and effective.
>
> We have addressed the feedback and have thoroughly revised the paper as follows.
>
> **Computational time:** We have added an analysis to Section 5.
>
> **Extension of method to more than 2 prompts**: We have added some results to the supplementary pdf, concept blending beyond three prompts is generally more useful in personalization, complex prompt generation, compositionality applications. These are interesting and standalone research projects and warrant thorough exploration into using our black Scholes concept blending idea for improved performance there.
>
> **Ablation on Avg CLIP Score for strike price**: We have added explanations for the strike price in Section 4, and have added an ablation experiment in the appendix.
>
> **More details on experimental setup**: We have added the details to Section 5 and the appendix.
>
> **Multi-object results**: In the second set of our dataset, we are focusing exclusively on the multi-object scenario.
>
>
> **Style blending comparisons**: We include style blending experiments in set 4 of our dataset, and compare with other concept blending methods qualitatively as well as quantitatively. The comparison methods that have been referenced by the reviewer solver DIFFERENT problems, and do not address the problem of concept blending. For instance, StyleAligned aims to generate different and unique images with the same style, ZipLoRA does personalization of subject and style (or multi-concept personalization), Cross-Image … does structure personalization and expressive image… deals with complex prompt image generation.
>
> Our approach to concept blending has the potential to be used for further research in other diffusion tasks such as personalization, style and appearance transfer, complex prompt image generation, etc that the referenced papers address, as standalone research projects. Our current paper focuses on the problem of concept blending.
>
> **Experiment on T2I Bench**
>
> T2I Bench is a benchmark for compositionality. Our paper focuses on a distinct problem statement: the fundamental issue of concept blending, for which we propose a dedicated algorithm. The primary aim of this work is to develop this algorithm for concept blending; we do not address compositionality at this stage, as that would require a standalone research project. We believe that incorporating compositionality within this paper would compromise the depth of our current analysis due to page limitations and project constraints. However, we concur with the reviewer that exploring the application of our proposed concept blending to enhance compositionality and personalization is a promising avenue for future research.
>
> Moreover, we anticipate that our method will be beneficial in various diffusion applications, including compositionality, story generation, and both single and multi-concept personalization. It could also be extended to the video domain, each of which presents intriguing standalone research opportunities that warrant thorough exploration.
>
> **Order of prompts**: No, the ordering of prompts does not matter. This is because, at every step of diffusion denoising, the algorithm chooses the prompt that should be selected by computing the Black Scholes score wrt each prompt, and this process is agnostic to the ordering of the prompts.
>
>
> We hope these responses clarify your concerns. Thank you once again for your valuable feedback, and we look forward to further engaging discussions!

---

> > ### Author Response · Authors · 2024-11-25
> >
> > Dear Reviewer,
> >
> > We would like to sincerely thank you once again for the time and effort you have devoted to reviewing our paper. With the discussion period ending in two days, we would greatly appreciate it if you could review our rebuttal at your convenience, should your schedule permit!
> >
> > If you have any questions, require additional explanations, or see any areas that need improvement, please feel free to let us know, and we will address them immediately!
> >
> > We are truly grateful for the valuable feedback you have provided on our research. Additionally, considering the improved current version, we would be sincerely thankful if you could consider raising your review rating!!!
> >
> > Hope you have a great week! Thank you very much!!!
> >
> > With warm regards,
> > The Authors

---

> > > ### Author Response · Authors · 2024-11-29
> > >
> > > Dear Reviewer,
> > >
> > > We would like to sincerely thank you once again for your time and efforts in helping us strengthen the paper. We have addressed all your valuable feedback and have made changes to the paper accordingly.
> > >
> > > As the discussion period is ending soon, we would love to hear your feedback on any other questions we can answer to clarify any remaining concerns.
> > >
> > > We sincerely appreciate your encouragement of our research potential, and we would be deeply grateful if you could consider raising the review rating.
> > >
> > > We are truly grateful to have you review our paper, and deeply appreciate your support!
> > >
> > > Thanks,
> > > Authors

---

### Official Review · Reviewer_vLKe · 2024-11-05

**Soundness:** 2
**Presentation:** 2
**Contribution:** 2
**Rating:** 5
**Confidence:** 3

**Summary:**

This paper proposes an approach to enhance prompt mixing in diffusion models. It leverages the concepts from the Black-Scholes algorithm used in financial markets. The authors analyze the relationship between diffusion models and the Black-Scholes model. It uses the connect to develop a method to select optimal text prompts during the denoising process. The method aims to generate images that effectively blend multiple concepts without requiring additional training or data collection. Experimental results indicate that the proposed method outperforms baselines in both CLIP scores and qualitative comparisons across various settings.

**Strengths:**

1. Novel conceptual connection: The paper makes an interesting theoretical connection between diffusion models and the Black-Scholes algorithm. It provides a new perspective on prompt mixing.

2. No additional training: The proposed method is data-efficient and requires no additional training or fine-tuning of the underlying diffusion model.

3. Comprehensive evaluation: The authors test their approach across multiple experimental settings with varying complexity and provide both qualitative and quantitative comparisons.

**Weaknesses:**

1. Weak theoretical foundation: While the paper tries to connect diffusion models with Black-Scholes, the connection is inadequately justified. The authors fail to rigorously demonstrate why the financial markets in the Black-Scholes model should apply to image generation. The mapping of concepts like "strike price" and "risk-free rate" to the image generation domain seems not very straightforward and intuitive.

2. Limited evaluation metrics: The paper relies heavily on CLIP scores for quantitative evaluation. However, CLIP scores is not the optimal evaluation strategy due to their vulnerability to hallucination and bias issues. More robust evaluation metrics should be given to validate the proposed method. Besides, the improvement of CLIP score is also minimal.
.
3. Insufficient ablation studies: The ablation studies to justify the specific choices made in adapting the Black-Scholes model are not given. For instance, the authors set the strike price K to a constant value of 0.25 without exploring how different values might affect the results. Similarly, the choice of risk-free rate as 1/T is not adequately explained or validated through experiments.

**Questions:**

1. Could the authors provide a more rigorous mathematical justification for why the market behavior should apply to the image generation domain?
2. What is the computational cost of computing Black-Scholes scores at each step?
3. How is the result with more than two prompts?

---

> ### Author Response · Authors · 2024-11-22
> **Response to reviews**
>
> We thank the reviewer for their time and valuable feedback. We are elated that the reviewer found our method to have a **novel conceptual connection** providing a new perspective, and **data efficient**. We are glad about the acknowledgement of our **extensive experimentation**.
>
> We have addressed the feedback and have thoroughly revised the paper as follows.
>
> **Theoretical foundation**: We have incorporated the feedback and enhanced Section 3, as well as the supplementary materials, with more robust analysis.
>
> **Limited evaluation metrics**: We have added three more evaluation metrics, BLIP, DINO and KID which help us assess overall text alignment with the combined text prompts, the preservation of attributes related to individual concepts, and the quality of the generated images. The Black Scholes algorithm for prompt mixing in diffusion models achieves superior results, compared to other prompt-mixing techniques, as also evidenced by the qualitative results.
>
> **Insufficient ablation studies**: We have added explanations for risk free rate and strike price in Section 4, and have added an ablation experiment on the strike price in the appendix.
>
> We hope these responses clarify your concerns. Thank you once again for your valuable feedback, and we look forward to further engaging discussions!

---

> > ### Author Response · Authors · 2024-11-25
> >
> > Dear Reviewer,
> >
> > We would like to sincerely thank you once again for the time and effort you have devoted to reviewing our paper. With the discussion period ending in two days, we would greatly appreciate it if you could review our rebuttal at your convenience, should your schedule permit!
> >
> > If you have any questions, require additional explanations, or see any areas that need improvement, please feel free to let us know, and we will address them immediately!
> >
> > We are truly grateful for the valuable feedback you have provided on our research. Additionally, considering the improved current version, we would be sincerely thankful if you could consider raising your review rating!!!
> >
> > Hope you have a great week! Thank you very much!!!
> >
> > With warm regards,
> > The Authors

---

> > > ### Author Response · Authors · 2024-11-29
> > >
> > > Dear Reviewer,
> > >
> > > We would like to sincerely thank you once again for your time and efforts in helping us strengthen the paper. We have addressed all your valuable feedback and have made changes to the paper accordingly.
> > >
> > > As the discussion period is ending soon, we would love to hear your feedback on any other questions we can answer to clarify any remaining concerns.
> > >
> > > We sincerely appreciate your encouragement of our research potential, and we would be deeply grateful if you could consider raising the review rating.
> > >
> > > We are truly grateful to have you review our paper, and deeply appreciate your support!
> > >
> > > Thanks,
> > > Authors

---

### Author Response · Authors · 2024-11-22
**Response to reviews**

We extend our heartfelt thanks to all the reviewers for their time and insightful feedback.

We are delighted that the reviewers found our method to be novel, interesting, and effective, with validation through thorough analysis.

We have addressed the reviewers' feedback and made extensive revisions to strengthen the paper. We thank the reviewers once again for their valuable feedback and look forward to engaging discussions.

---

### Meta-Review · Area_Chair_ge9N · 2024-12-20

**Metareview:**

This paper introduces the Black-Scholes model for pricing options in finance into diffusion models for prompt mixing. After the rebuttal discussion, most reviewers are still not fully convinced in the motivation and justification of the method, and the experimental performance. While the connection between diffusion models and the Black-Scholes model is novel and interesting conceptually, AC agrees that this work needs further improvement in the presentation, the justification of the method, and the evaluation on more challenging prompt mixing and compositionality.

**Additional Comments On Reviewer Discussion:**

Two main concerns raised by the reviewers:

1. the lack of a strong motivation and justification for the method
2. the experiments are limited: no systematic evaluation on compositionality and the mixing is limited to two prompts.

The authors provided some justification arguments in the revised draft for issue 1 and some new preliminary experimental results on the mixing of three concepts for issue 2. After the rebuttal, the reviewers are still not fully convinced.  I tend to agree with the reviewers that further insights and formal analysis are necessary to more clearly understand the connection between the Black-Scholes model  and the diffusion models, and more critically, a systematic evaluation on more challenging prompt mixing and compositionality will greatly improve the quality of this submission.

---

### Decision · Program_Chairs · 2025-01-22

Reject